# Preparation and Complex Characterisation of Stabilised Gold Nanoparticles: Biodistribution and Application for High Resolution In Vivo Imaging

**DOI:** 10.3390/ph17111479

**Published:** 2024-11-03

**Authors:** Jaroslav Turánek, Pavlína Turánek Knötigová, Pavel Kulich, Radim Skoupý, Kamila Hrubanová, Naděžda Vaškovicová, Ladislav Fekete, Antonín Kaňa, Robert Mikulík, Milan Raška

**Affiliations:** 1Neurology Department, The International Clinical Research Center ICRC of St. Anne’s University Hospital in Brno, Pekařská 53, 656 91 Brno, Czech Republic; knotigova.p@seznam.cz (P.T.K.); robert.mikulik@fnusa.cz (R.M.); 2Department of Immunology, Faculty of Medicine and Dentistry, Palacky University Olomouc, Hněvotínská 3, 775 15 Olomouc, Czech Republic; milan.raska@upol.cz; 3Institute of Clinical Immunology and Allergology, University Hospital Hradec Kralove and Faculty of Medicine in Hradec Kralove, Charles University, Sokolská 581, 50005 Hradec Kralove, Czech Republic; 4Veterinary Research Institute, v.v.i., Hudcova 296/70, 621 00 Brno, Czech Republic; kulich.pavel@seznam.cz; 5Institute of Scientific Instruments, v.v.i., AS CR, Královopolská 147, 612 00 Brno, Czech Republic; ras@isibrno.cz (R.S.); dobranska@isibrno.cz (K.H.); 6Department of Histology and Embryology, Faculty of Medicine, Masaryk University, Kamenice 753/5, 62500 Brno, Czech Republic; nadezda.vaskovicova@med.muni.cz; 7Institute of Physics, Czech Academy of Sciences, Na Slovance 2, 18200 Prague 8, Czech Republic; fekete@fzu.cz; 8Department of Analytical Chemistry, University of Chemistry and Technology, Prague, Technická 5, 166 28 Praha 6, Czech Republic; antonin.kana@vscht.cz

**Keywords:** gold nanoparticles, in vivo imaging, microcomputer tomography, biodistribution, nanotoxicology

## Abstract

The Turkevich method was optimized to prepare gold nanoparticles (AuNP) stabilized by polyethyleneglycol (PEG) for µCT. Using various independent modalities, we thoroughly characterized the optimized PEG-AuNPs. Here, we show that PEG-AuNPs are retained in the blood and provide a high contrast in the high-resolution µCT imaging of blood vessels and inner organs. The biodistribution is characterized by prolonged circulation in the blood and accumulation in the liver, spleen and skin. The accumulation of AuNP in the skin resulted in the blue discoloration of eyes and the whole skin. In vitro experiments using a leukemic monocyte THP-1 cell line model expressing high levels of NLRP3 demonstrated that the NLRP3inflammasome was not activated by PEG AuNP. Over 9 months, the mice were scanned by µCT and were in good health. Scans in mice using PEG-stabilized AuNPs in this study were sharper, with a higher contrast, when compared to a commercial contrasting agent at the same dose. The PEG-AuNPs were morphologically and chemically stable for at least two years when stored in the refrigerator.

## 1. Introduction 

The application of AuNPs in preclinical in vivo imaging (e.g., computed tomography, photothermal imaging, surface-enhanced Raman scattering, photoacoustic imaging) requires these particles to provide high-quality contrast and not affect physiological functions in the short or long term. Medical imaging provides key information for the early detection, diagnosis, and appropriate treatment of diseases. Gold nanoparticles (AuNPs) have become increasingly significant in medical imaging due to their unique properties and potential for enhancing diagnostic techniques. The interactions of AuNPs with biological systems are tightly linked to their physiochemical properties. These include the nanoparticle size and shape (spherical nanoparticle, nanodiscs, rods, cubic), surface modification (e.g., polymer coating, conjugation with targeting ligands), and internal structure (crystallinity, hybrid nanoparticles). Variations in any of these parameters could be toxic to the recipients of AuNPs. To make gold nanoparticles more suitable for in vivo applications (reduce aggregation, enhances solubility), the coating of gold nanoparticles with polyethylene glycol (PEG) is used [1,2].

X-ray-based imaging techniques are the most widely utilized medical imaging technologies due to their high spatial resolution, fast image acquisition, and low cost. These imaging techniques rely on ionizing electromagnetic radiation to generate contrast based on the differential X-ray attenuation properties of the tissues. X-ray high-resolution micro-CT (µCT) is a phase-contrast and darkfield CT used in preclinical research for imaging small animals [3,4].

X-ray micro-CT has several unique properties such as the strong and tunable attenuation of light, fluorescence, easy light-to-heat conversion, and the attenuation of gold nanoparticles. This imaging technique could be utilized in a broad range of biomedical applications like computed tomography, photoacoustic imaging, magnetic resonance imaging or near-infrared radiation (NIR)-induced hyperthermia [5,6,7,8].

Several methods for gold nanoparticle (AuNPs) preparation have been proposed. The plethora of methods in the literature suggests that gold nanoparticles of different shapes, sizes, structures, and with surface modifications are formed. These factors affect the nanoparticle’s quality of contrast, physical and chemical stability, tissue and organ targeting, biodistribution, and elimination [9]. The Turkevich, Frens, and Brust–Shifrin methods, which produce quasi-spherical AuNPs, are the most widely utilized, in part due to their simplicity and scalability [10,11]. Notably, the structure, size distribution, and ƺ-potential of gold nanoparticles are crucial parameters related to their long-term stability, role in organisms, and biodistribution. Also, their ability to act as a contrast for in vivo imaging and as a theranostic element is dependent on their physical parameters and ƺ-potential. Here, we describe the optimized preparation and characterization of highly stable PEG-modified gold nanoparticles. Using a murine model, we demonstrate the precise high-resolution imaging of the mouse vascular system. The main novelty of this work lies in a comprehensive characterization of laboratory-prepared gold nanoparticles and their use in high-resolution imaging of the cardiovascular system in small laboratory animals using the high-resolution µCT.

## 2. Material and Methods

### 2.1. Preparation of PEG-AuNPs

The PEGylated gold nanoparticles (PEG-AuNPs) used in the in vivo experiments were prepared as described previously, with some modification [12].

Briefly, citrate-stabilized Au-NPs were synthesized and a tenfold molar excess of trisodium citrate was added to a boiling solution of 0.5 mM chloroauric acid with vigorous stirring. The reaction conditions were kept unchanged until the color of the mixture turned ruby red. The citrate-stabilized AuNPs were PEGylated via the addition of methoxy-PEG-thiol 2 kDa. In total, 100 nmol/mg (mPEG-SH/AuNPs) was used. After PEG addition, the colloid solution was incubated at room temperature overnight with stirring.

The PEG-AuNPs solution was concentrated using tangential flow filtration (TFF KrosFlo KR2i; HF Filter: MIDIKROS P/N: D04-E030-05-N; Repligen, Boston, MS, USA). Thereafter, the concentrated colloid solution of PEG-AuNPs was sterilized by dead-end filtration (TPP Syringe-Filter P/N: 99722). The final concentration was reached through centrifugation (50,000× *g*/90 min).

The surface modification of colloidal gold using the chemisorption of alkanethiols in the presence of a nonionic surfactant was performed as described previously [13].

### 2.2. Transmission Electron Microscopy

The measurement and morphological analysis of the Au nanoparticles were performed by transmission electron microscopy 268D Morgagni TEM (Thermo Fisher Scientific, Eindhoven, The Netherlands). 

A citrate-stabilized or pegylated AuNPs suspension (0, 1 mg/mL) was deposited on the surface of a Formvar-coated 300 mesh copper grid (300 Old Mesh, Agar Scientific, Vienna, Austria). After a 30 s incubation, excess fluid was removed using filter paper. TEM observations were obtained at 90 kV and a magnification of 180,000–180,000×. In total, 200 particles were used to measure the size distribution.

### 2.3. Ultrathin Sections

The measurement and morphological analysis of the Au nanoparticles were performed by transmission electron microscopy (Philips EM 208 and 268D Morgagni TEM) (FEI, Brno, Czech Republic, and Thermo Fisher Scientific, Eindhoven, The Netherlands). 

A citrate-stabilized or pegylated AuNPs suspension (0, 1 mg/mL) was deposited on the surface of a Formvar-coated 300 mesh copper grid (300 Old Mesh, Agar Scientific, Biedermannsdorf, Austria). After a 30 s incubation, excess fluid was removed using filter paper. TEM observations were obtained at 90 kV and a magnification of 180,000–180,000× . In total, 200 particles were used to measure the size distribution.

### 2.4. Scanning Electron Microscopy and Spectral Analysis

An EDX spectroscopy system was used to analyze the spleen and skin energy dispersed in STEM mode. 

A Magellan 400 SEM (FEI, Brno, Czech Republic) equipped with an energy-dispersive spectroscopy detector octane elect super (EDAX, Weiterstadt, Germany) was used for the SEM-EDX analysis. All measurements were taken at an acceleration voltage of 30 kV and a probe current of 400 pA. The tissue samples were mounted on Copper TEM grids and stainless steel sample holders to prevent background noise from the gold signal. A standardless algorithm incorporated into the EDX detector software calculated the amounts for each element.

### 2.5. Particle Size Analysis

The projected area of individual particles was visualized by electron microscopy and 10 particles were analyzed using ImageJ software. To convert the area to particle diameter, all examined particles were assumed to be spherical. The particle diameter was derived using the following formula:D = 2 × √(A/π)(1)
where D is the particle diameter and A is the projected area.

Finally, the mean particle diameters were calculated. A frequency polygon was constructed using the same bin sizes used for DLS measurements.

### 2.6. Characterization of Nanoparticles by Light Scattering

The AuNP size characterization, ζ-potential and particle concentration estimates (particles/mL) were measured using a Zetasizer Utra Plus (Malvern Panalytical, Malvern, UK). The back scatter and multi-angle experiments were measured at 25 °C in different media (ZEN2112 cuvette). The ζ-potential of the PEG-AuNPs was measured at 25 °C in 10 mM of sodium phosphate buffer at pH 7.4 (DTS 1070 cuvette). The ζ-potential of unpurified citrate-stabilized AuNPs was measured at 25 °C to prevent rapid aggregation. 

### 2.7. Analysis of AuNPs by Nanoflow Cytometry (nanoFCM)

A NanoAnalyzer (nanoFCM, GB) equipped with a 488 nm laser was used for analyses of the AuNPs. Nanoflow cytometry uses single-photon detectors (SPCM) to measure scattered and refracted light from sub-micron particles. The intensity of side scattered light was compared to a standard curve generated by measuring the diameters of 10 nm, 20 nm, 40 nm nanogold standards (Merck, Rahway, NJ, USA). The sample concentration was determined by comparing the AuNPs to 250 nm silica nanoparticles of a known particle concentration (NanoFCM, Xiamen, China) [14]. 

The AuNPs were diluted 150× using Milli–Q water. The diluted suspension was filtered using a 0.2 μm filter and measured on a nanoflow instrument. The laser was set to 15 mW and 100% SS decay. Measurements were taken over 1 min at a sampling pressure of 1.0 kPa. The data were generated and analyzed in the NanoFCM Professional Suite v1.8 software.

### 2.8. Atomic Force Microscopy of PEG Stabilized Au-NP

Atomic force microscopy (AFM) using an ambient AFM (Bruker Optics, Brno, Czech Republic) in peak force tapping mode with ScanAsyst air tips (Bruker; k = 0.4 N/m; nominal tip radius 2 nm) was carried out at room temperature. The topographies of 512 × 512 points^2^ resolution were measured. Water-diluted suspensions were dropped on a polished silicon substrate. After the water had evaporated, several 1 × 1 micron^2^ particle images were recorded. The mean height of the particles was evaluated using a single Gaussian fit of the histogram from 505 particles (bin size 0.4 nm).

### 2.9. FTIR Spectroscopy 

The infrared spectra of the AuNPs were measured using an Invenio S (Bruker Optics, Brno, Czech Republic) infrared spectrometer. At 25 °C, dry PEG-AuNPs in a diamond cell were measured at a resolution set at 4 cm^−1^ for 120 scans with a wave number range of 3000–450 cm^−1^.

### 2.10. Single-Particle ICP-MS Analysis

Gold nanoparticles standards (Merck, USA) were sonicated (Elmasonic ultrasonic bath S40, Elma, Einigen, Germany) for 10 min. In total, 450 µL of the AuNP standard was mixed with 50 µL of 1% Tween 20 (*v*/*v*) solution for nanoparticle stabilization. After 30 min, the mixture was diluted 1,000,000-fold and immediately analyzed using ICP-MS in a single-particle mode (sp-ICP-MS, PerkinElmer NexION 350D, Concord, ON, Canada). The sample was prepared and analyzed in the same way; only the Tween 20 step was omitted. Data were captured and analyzed using a Syngistix 1.1 Nano module (Perkin Elmer). The sp-ICP-MS conditions were set as follows: RF power of 1.1 kW, nebulizer gas flow rate of 0.74 L/min, auxiliary gas flow rate of 1 L/min, plasma gas flow rate of 11 L/min, sample uptake of 0.33 mL/ min, dwell time of 0.1 ms, acquisition time of 60 s, and 197Au isotope. The sp-ICP-MS measurements were calibrated using the above-mentioned AuNP standards. Triplicate samples were measured for the Au signal intensities, number of signals, diameters and concentrations of AuNPs (for detail, see Loula et al., 2019) [15].

### 2.11. UV-Vis Measurements

To obtain the UV-Vis spectra, a Specord 600 instrument was used (Zeiss, Jena, Germany). The spectrum was acquired within a wavelength of 350–800 nm with a 1 nm resolution. The spectra were used to evaluate the relative level of particle aggregation (flocculation parameter). The integral of the absorption (normalized) spectra within the 600–800 nm wavelength defined the flocculation parameter [13,16].

### 2.12. Serum Stability Test

The aggregation stability of the PEG-AuNPs was tested in a physiologically relevant media of 80% fetal bovine serum (FBS) for 10 h. Because proteins in FBS can form aggregates, light-scattering methods like DLS are not appropriate. Therefore, UV-Vis was used to monitor the changes in the aggregation of the nanoparticles. The flocculation parameter was used to evaluate the time evolution suspension stability.

Sample preparation: purified PEG-AuNPs in MiliQ water (0.1 mg/mL) were mixed with FBS at a 1:4 ratio. After mixing, the absorbance was measured immediately and at 60 min intervals for 10 h. The entire experiment was conducted at 37 °C.

### 2.13. In Vitro Testing of Gold Nanoparticles Biocompatibility—Activation of Inflammasome 

A cell-based assay using THP-1-null cells expressing high levels of NLRP3, adaptor protein ASC (apoptosis-associated Speck-like protein with a caspase recruitment domain), and pro-caspase 1 together with reporter HEK-Blue™ IL-1β (Invivogen, Paris, France) was applied as previously described [17,18].

### 2.14. Viability Testing

Non-adherent THP-1 and adherent B16F1 cells were used for viability testing using the manufacturer’s protocol (Tali^TM^ Apoptosis Kit-Annexin V Alexa Fluor™ 488 & Propidium Iodide (PI), Life Technologies Corporation, Thermo Fisher Scientific, Prague, Czech Republic). The cells were incubated with 100, 10 and 1 µg/mL concentrations of PEG-AuNP nanoparticles for 24 and 48 h. Negative control cells were incubated without the apoptosis-inducing agent. To remove cell autofluorescence, a second negative sample of unstained cells was prepared. The cells were rinsed, and thereafter incubated with Annexin V Alexa Fluor^®^ 488 and propidium iodide according to procedures recommended by the manufacturer. A flow cytometer BD FACSymphony A1 cell analyzer (BD, Prague, Czech Republic) was used to evaluate the cell viability.

### 2.15. In Vivo µCT Imaging of Mice

The mouse model was scanned on a MicroCT SkyScan 1276 (Bruker, Ettlingen, Germany). Single images were acquired by continuous rotational scanning with a low-dose filter. Each scan consisted of 4 parts that were combined during computer reconstruction into a single image in InstaRecon^®^. CTvox and CTAn were used for 3D visualization and morphometry. The mouse model was scanned under 2.5% isoflurane general inhalational anesthesia (Isoflurin 1000 mg/g inh liq 250 mL (isoflurane)) [SW1] in an R540 Mice & Rat anesthesia machine (USA: 6540 Lusk Blvd., Suite C161 San Diego, CA, USA). Animal experiments were approved by the Ethic committee of the Ministry of Agriculture of the Czech Republic under serial number MZe 2008.

## 3. Results and Discussion

### 3.1. Preparation and Characterisation of AuNP

The TEM analysis revealed almost spherical Au nanoparticles (Figure 1A). During the two-year storage in the refrigerator at 4 °C, no significant changes in morphology and size were observed. The Au core diameter was 14.48 ± 1.1 nm (most particles were found in the 14.0–14.9 nm range) (Figure 1B). The TEM analysis of the prepared AuNPs showed a high degree of homogeneity, without crystalline domains, and a low mass density comparable to commercial products [19,20]. Notably, mass density inhomogeneity can decrease the contrast in µCT imaging.

### 3.2. ICP-MS Analysis

In contrast to DLS, single-particle ICP-MS analysis individually measures each gold nanoparticle irrespective of its surface modification. The ICP-MS analysis showed an AuNP core mean diameter of 14.7 ± 0.6 nm. These data perfectly correlate with the TEM analysis. Also, when the number of particles measured by ICP-MS (1.86 ± 0.06 × 10^11^ mL^−1^) was compared to MADLS (1.55 ± 0.09 × 10^11^ mL^−1^), minimal differences were noted (Figure 2 and Figure 3). The ICP-MS analysis confirms a high degree of homogeneity, without microcrystalline domains and domains with a low mass density.

### 3.3. Analysis by Nano Flow Cytometry (nanoFCM)

A NanoFlow analyzer was used to accurately measure the size distribution and number of AuNPs. A mean diameter of 14.7 ± 1.3 nm was obtained. Although this was identical to the data from TEM and ICP-MS, the size of the AuNPs was calculated from the flat lower part of the calibration curve (Figure 3). Most AuNPs (96%) are clustered between 12 and 21 nm and only 4% larger in size (Figure 3). Although nanoflow cytometry can determine the number of particles, a suitable AuNP standard was not available; therefore, this measurement was not performed.

### 3.4. Measurement of Size Distribution, ζ-Potential and Number of AuNP Counting by MADLS

PEG-AuNPs are a structural hybrid of nanoparticles containing an Au hard core and a soft low-density PEG corona. The multiangle dynamic light-scattering method (MADLS) was used to obtain the real dimension of the PEG-AuNP particle (Au core and PEG corona) expressed as the hydrodynamic radius (R_h_). As expected, the hydrodynamic diameter of the prepared PEG-AuNP was higher than that of the Au core, and the data are summarized in Figure 4. We obtained a mean hydrodynamic diameter from light scattering of 22.6 ± 0.9 nm (distribution expressed as the number of nanoparticles), and for larger particles (presumably aggregates) above 100 nm, the nanoparticle concentration was below 1 ppm. A PEG corona thickness of approximately 4 nm and a molecular weight of 2000 Da were observed. This is identical to the length of stretched PEG molecules. Moreover, it confirms the brush structure of the PEG corona on the AuNP surface. Interestingly, no signs of aggregation or morphological changes were observed in the 80% fetal bovine serum for 10 h at 37 °C. 

A fraction of particles above 100 nm was detected by MADLS (Figure 4A size distribution according to intensities), which corresponds to the results of the nanoflow cytometry.

The ζ-potential of the citrate-stabilized AuNPs was −36.2 ± 1.2 mV. This implies that a negatively charged citrate molecule is bound to the surface of AuNPs. The ζ-potential of the PEG-stabilized AuNPs was −2.3 ± 0.4 mV. The change from a negative ζ-potential to neutral reflects the surface citrate displacement and binding of PEG via an Au-S-PEG bond (Figure 4). The formation of the PEG corona is demonstrated by an increase in the hydrodynamic radius and the nearly neutral value of the ζ-potential. 

### 3.5. UV-VIS Spectroscopy

UV-VIS spectroscopy is a simple method used to measure the size and concentration of AuNPs. An AuNP size of 14.2 ± 0.85 nm was observed. The concentration of AuNPs was estimated as 1.88 ± 0.31 × 10^11^. The results perfectly correlate with the data from ICP-MS, MADLS, the nanoFlow Analyzer, and TEM. The number of particles counted by ICP-MS, MADLS and UV-Vis are also identical (Figure 5).

### 3.6. Infrared Spectroscopy 

The analysis of PEG-AuNPs by FTIR confirmed the surface modification of the AuNPs by PEG. Vibration bends belonging to chemical bands in PEG were found in the FTIR spectrum (Figure 6). Although a weak S-Au signal was measured, it confirms the covalent binding of PEG to the surface of AuNPs. To visualize the weak S-Au signal, a vibration at 668 cm^−1^ was performed with a dry PEG-AuNP sample in a diamond cell. The vibration of the Au-S bond at 668 cm^−1^ is similar to the FTIR spectrum published by Omar et al. [21]. Characteristic vibration bands confirming the covalent PEG modification of the AuNPs were present in the FTIR spectrum. This concurs with data obtained by dynamic light scattering regarding the increase in R_h_ (Figure 4) and confirms the PEG modification of the AuNP surface.

### 3.7. Atomic Force Microscopy of PEG Stabilized Au-NP

AFM was used for the detailed analysis of the AuNP size distribution. Figure 4 shows the size (particle height) distribution of AuNP particles [22]. AFM analysis showed the normal size distribution of the AuNP particles. The average AuNP particle height was 11.7 nm (Figure 7). This slight underestimation of the Au particle height by the AFM method is due to the tip parameters and specific properties of the PEG corona.

### 3.8. Stability of AuNP in Foetal Bovine Serum

The incubation of PEG-AuNPs with MC at 37 °C did not change the UV-Vis spectrum of the PEG-AuNPs. This implies that the nanoparticles are stable and will not aggregate in the bloodstream. Therefore, absorption by macrophages in the liver, lungs, spleen, and other organs is delayed. Citrate-stabilized AuNPs interacting with FBS were quickly replaced by serum proteins, as shown in the UV-Vis spectrum data summarized in Appendix A.

### 3.9. In Vitro Activation of Inflammasome NLRP-3 and Intracellular Distribution in THP-1 Cells

An In vitro model of NLRP3 expressing the THP-1 cell line was used to test the potential NLRP3-IL-1β-mediated proinflammatory activity of the PEG-AuNPs. No IL-1β secretion was observed within the concentration range of 5–1000 µg/mL (Figure 8). Also, the incubation of cells with PEG-AuNPs plus LPS does not significantly increase IL-1β secretion. We show that the PEG-AuNPs did not activate the NLRP3 terminal pathway responsible for IL-1β caspase-mediated cleavage and secretion [23]. The detailed effect of the PEG-AuNP size on the inflammatory response indicates the preferential activation of NLRP3-dependent IL-1β production by small-sized PEG-AuNPs (4.5 nm). Particles > 10 nm, including those analyzed here, activate IL-6 and TNF-α in an NF-κB-dependent manner [22]. THP-1 cells incorporate AuNPs by a mechanism of micropinocytosis and the accumulation of AuNPs in intracellular vesicles, as revealed by TEM (Figure 9). In vitro testing using non-adherent THP-1 cells and adherent B16F1 cells for two days showed that no signs of cytotoxicity were induced by PEG-AuNP within a range of 1–100 µg/mL (Appendix A). A comparison of the Annexin V and PI staining of the cells shows that no significant differences in the percentage of apoptotic and necrotic cells were recorded among PEG-AuNP-treated and untreated control cells irrespective of the AuNP concentration tested (Appendix A) [24]. The cytotoxicity of PEG-AuNPs depends on several factors, including the method of PEGylation and the presence of various functional groups affecting the AuNP surface charge, which influences the interaction with the cell membrane, endocytosis, and toxicity [25]. The identification of black spots as AuNPs was proven by an EDX detector (data in Appendix A).

### 3.10. Application of AuNP for µCT In Vivo Imaging

We compared stabilized PEG-AuNPs with the AuroVist™ 15 nm gold blood Pool X-ray contrast agent, a commercially available nanogold contrast agent. Using the recommended dose of AuroVist™ 15 nm for 20 mg/mice (1g/kg) for the high-contrast µCT visualization of blood vessels, PEG-AuNPs were applied to mice. Notably, the PEG-stabilized AuNPs yielded a stronger contrast at the same concentration (Figure 10).

An animal-friendly “low-dose” filter was used during repeated mouse scanning. Interestingly, although fast rotational methods ensure that animals are not scanned over long periods with high radiation doses, the images obtained using this procedure are of low resolution. Nevertheless, we observed the longer blood retention of PEG-AuNPs.

It is surprising that gold nanoparticles of similar size and size distribution differed so much in the contrast provided. The reason for this may be their half-life in the bloodstream due to the stability of their surface coverage. Also, the internal structure of their core, determined by the method and execution of the preparation, can have an effect on the contrast intensity. Our experimental work was not focused on this direction, and to clarify the influence of the internal structure of gold nanoparticles on the contrast intensity, a new grant project needs to be prepared.

### 3.11. Biodistribution of AuNP 

The skin of the mice treated with AuNPs turned dark blue shortly after the intra venous application of AuNPs. This color persisted for the 9-month duration of the experiment (Figure 11). Chrysiasis, or the hyperpigmentation caused by parenteral gold therapy, was first described in 1928 when gold therapy was widely used for several common conditions such as tuberculosis [26].

### 3.12. Deposition of AuNP in Various Organs

The AuNPs accumulated in the spleen and liver (Figure 12). The PEG-AuNPs were cleaned from the blood circulation by macrophages. Presumably, the AuNPs accumulated in the cytoplasm, and interacted with vesicles and other organelles. The penetration of AuNPs into the nucleus was negligible. However, some AuNPs seem to be associated with the nuclear membrane (Figure 9). TEM with an EDX detector showed that the black spots are nanogold particles (Appendix A).

AuNP deposition in the spleen and hepatocytes was shown by EDX analyses of spots. Examples of these analyses are presented in the Appendix A.

The dynamics of AuNP biodistribution were analyzed by µCT and expressed as Hounsfield units (HU) (Figure 13). Visualization of the blood vessels and the heart was performed 24 h after the administration of AuNPs. The graph shows the accumulation of AuNPs in the heart, spleen and liver.

Deposition of the tested gold nanoparticles in the skin and other internal organs had no observable toxic effect on mice. This is consistent with the inertness of gold nanoparticles and their low pro-inflammatory potential, as shown by in vitro experiments. Nevertheless, the accumulation of gold nanoparticles (larger than 8 nm) should be considered as a possible source of interference in pharmacological experiments where computed microtomography is used.

## 4. Conclusions

In this work, we compared different methods for the complex characterization of AuNPs prepared by an optimized Turkevich method. We used these PEG-stabilized AuNPs for the imaging of the cardiovascular system in small laboratory animals using high-resolution microcomputed tomography and the results were compared to the commercial preparation AuroVist™ 15 nm. Several studies have demonstrated the in vivo and in vitro toxicity of various AuNPs due to modifications in the physiochemical parameters [27,28]. Although our study was not focused on the toxicology of AuNPS, we show the inertness of our optimized PEG-AuNPs. Furthermore, the NLRP3 inflammasome, an important sensor of danger signals, was not activated.

The main novelty of this work lies in its comprehensive characterization of laboratory-prepared gold nanoparticles and their use in the high-resolution imaging of the cardiovascular system of small laboratory animals using high-resolution µCT. For µCT imaging, it is important that PEG-AuNP application did not interfere with repeated anesthesia when scanning was performed over a long period. Our mice models were in perfect health even after repeated µCT scanning in the 9-month period. The µCT demonstrated the accumulation of PEG-AuNPs in the spleen and liver. The observed chrysiasis confirms the accumulation in the skin and eyes. Stabilized AuNP preparations are being investigated for various biomedical applications. Their application as functional hydrophilic biocompatible polymers could be exploited in theranostics and in the rapid visualization of thrombi in preclinical animal models.

## Figures and Tables

**Figure 1 pharmaceuticals-17-01479-f001:**
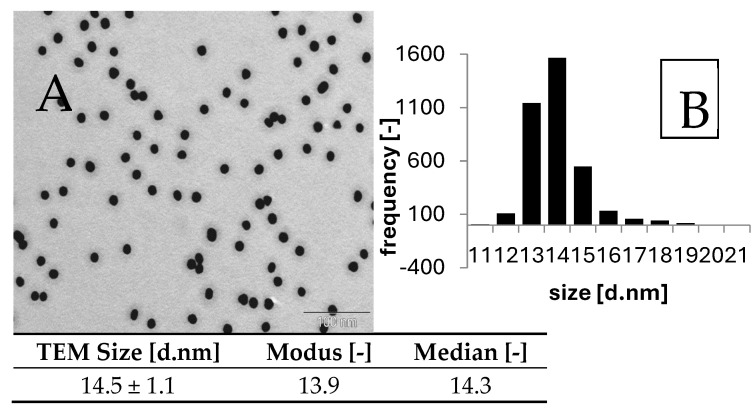
**TEM photograph of AuNP** (**A**) stabilized by PEG-SH; (**B**) histogram of size distribution calculated from TEM.

**Figure 2 pharmaceuticals-17-01479-f002:**
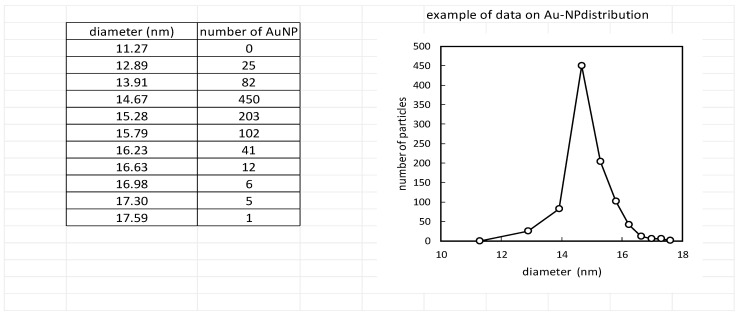
The size distribution of AuNPs measured by ICP-MS.

**Figure 3 pharmaceuticals-17-01479-f003:**
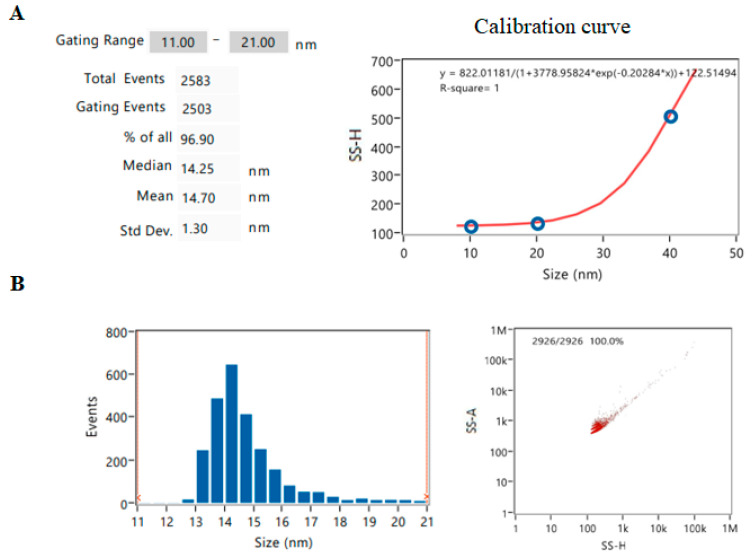
Analysis of AuNP by nanoFlow analyzer. (**A**) Results from size analysis and calibration curve of AuNP standards with sizes of 10, 20 and 40 nm in diameter. (**B**) Histogram of size distribution and dot plot of AuNPs.

**Figure 4 pharmaceuticals-17-01479-f004:**
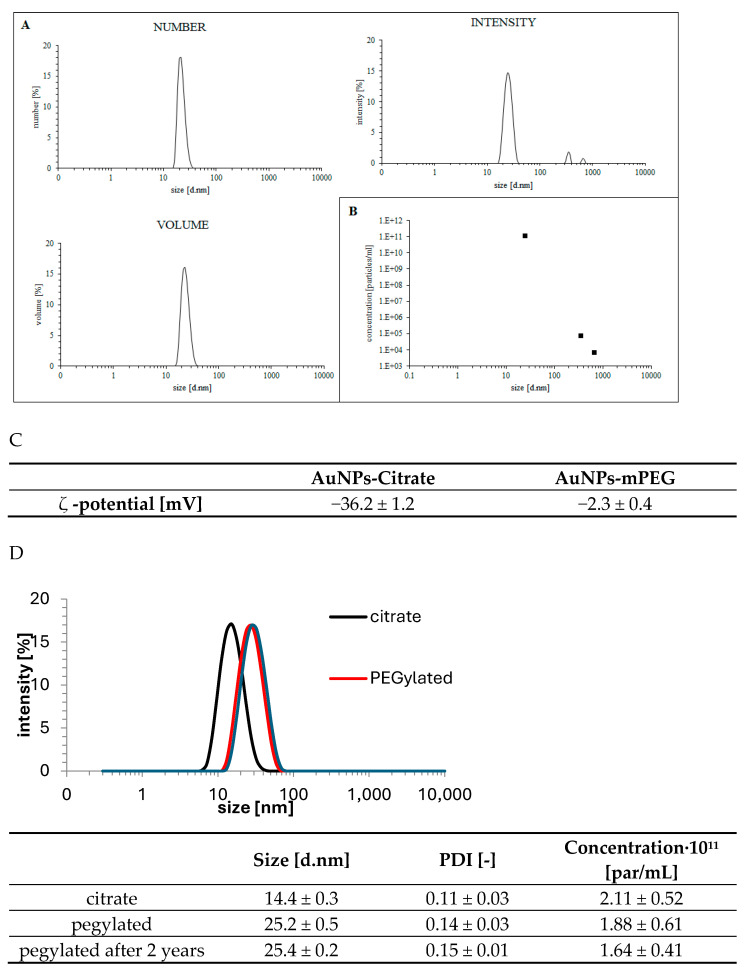
Analyses of AuNP by MADLS: The size distribution is expressed as the number, intensity and volume percentage. (**A**) The number, intensity and volume of nanoparticles of a specific size. (**B**) The size and concentration of nanoparticles calculated from MADLS data. (**C**) ζ-potential of citrate-stabilized AuNPs and ζ-potential of PEG-stabilized AuNPs. (**D**) Analyses of fresh (red curve) and PEG-stabilized AuNPs prepared by dynamic light scattering and stored for two years (grey curve). The table displays the number of particles in various groups, as calculated using the intensity distribution obtained by MADLS. AuNPs stabilized by citrate are shown as the control (black curve). The size distribution is expressed as the intensity percentage.

**Figure 5 pharmaceuticals-17-01479-f005:**
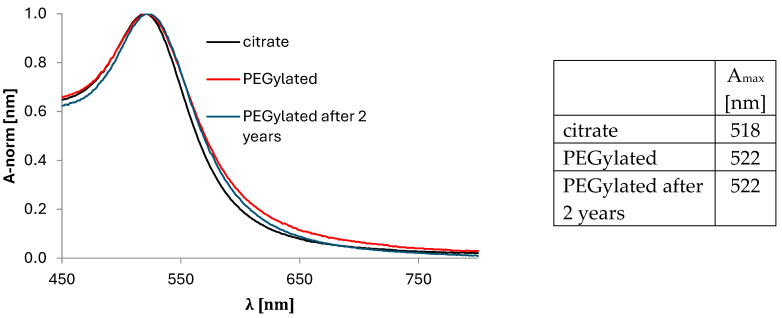
Normalized UV-Vis spectra of AuNPs stabilized by citrate and PEG, and spectrum of Au-NPs stabilized by PEG and stored for two years in a refrigerator.

**Figure 6 pharmaceuticals-17-01479-f006:**
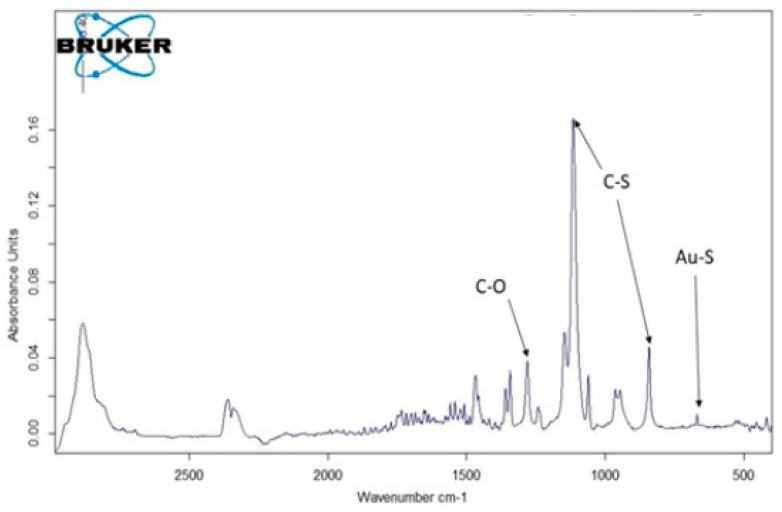
FTIR spectrum of PEG-AuNPs. The vibration of C-O, C-S and S-Au bonds is marked. 842.51 cm^−1^ correspond to C-O vibration, 1115.53 cm^−1^ and 1279.76 cm^−1^ correspond to C-S vibration and 668.33 cm^−1^ corresponds to Au-S vibration.

**Figure 7 pharmaceuticals-17-01479-f007:**
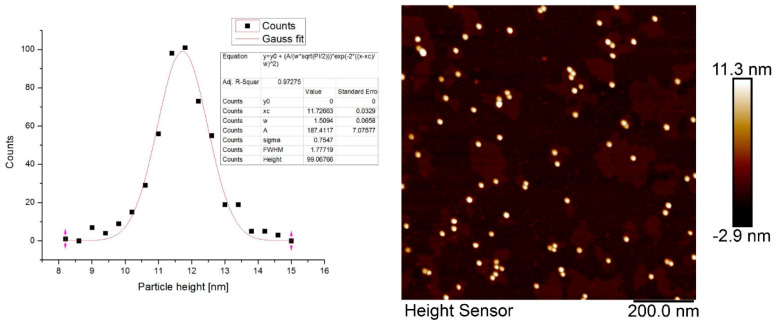
Characterization of AuNP particles’ size distribution. (**Left**): Particle height distribution of AuNP particles by AFM. (**Right**): Atomic force micrographs of the AuNP particles.

**Figure 8 pharmaceuticals-17-01479-f008:**
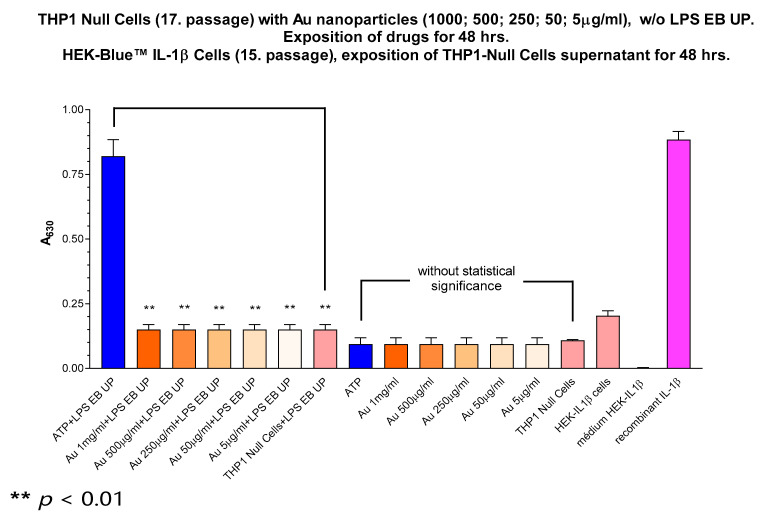
Inflammasome NLRP3 is not activated by PEG-AuNPs in vitro. THP-1 null cells were incubated with PEG-AuNPs (1000; 500; 250; 50; 5 µg/mL) for 48 h with LPS or without LPS. HEK- Blue IL-1β cells were exposed to the supernatant from THP-1 null cells for 48 h to determine the concentration of IL-1β released from activated THP-1 null cells. ATP and IL-1β were used as positive controls.

**Figure 9 pharmaceuticals-17-01479-f009:**
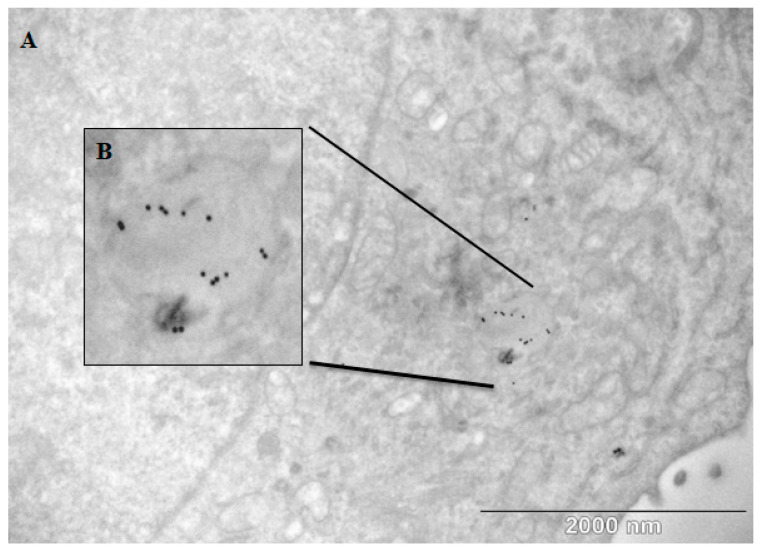
Deposition of gold nanoparticles in THP-1 cells exposed in vitro 48 h to AuNPs (10 µg/mL) visualized by TEM. AuNPs (black dots) were localized presumably inside intracellular vesicles (**A**). Detailed picture of AuNPs inside of the vesicle (**B**).

**Figure 10 pharmaceuticals-17-01479-f010:**
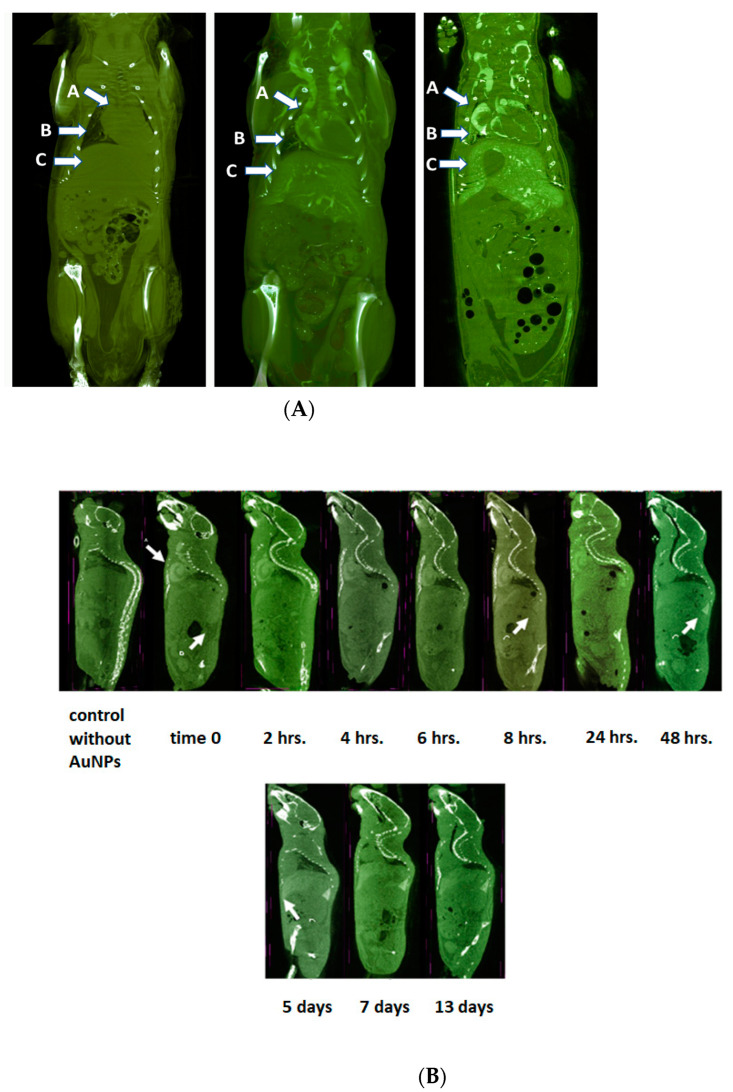
Comparison of in vivo contrast signals generated by Aurovist and stabilized AuNPs. Visualization of skeleton and vascular system of the mouse. (**A**) Untreated mice (**left**); mice treated by Aurovist 20 mg/mice (**middle**); mice treated by stabilized AuNPs at 20 mg/mice (**left**) A–heard, B–lungs, C–liver. Visualization of the heart chambers filled with blood with contrast material, B-visualization of the lung, C–visualization of the liver: the large vessels in the liver are clearly visible and the liver lobes begin to be displayed 2 h after application; this is due to the high flow of blood with imaging agent through this organ. After 48 h, they are fully imaged, as Au nanoparticles are taken up by the liver cells (macrophages). Magnification was 42.5 µm, binning 1K. Each scan consisted of four parts, which were combined into one image during computer reconstruction in the InstaRecon^®^ program. The 3D visualization and morphometry were performed in the program CTvox and CTAn. (**B**) µCT scans of mice after 0–8 h and 1–13 days (after the intravenous administration of 20 mg of AuNPs modified by mPEG). Representative scans of mice contrasted by intra venous application of AuNPs (20 mg/mice). Note the contrast image of the head between zero and 24 h. The contrast of the liver increases within 1–13 days. The contrast image of the spleen is visible after 48h (white arrow at 48 h). The scanning parameters are the same as described above. (**C**) Visualization of skeleton and vasculature in mice by µCT. A full video is presented in the Appendix A. The mice were anesthetized using RWD animal isoflurane anesthesia vaporizer for small animals and injected intra venously with PEG-AuNPs. Two hours after administration, a µCT scan was made. The µCT visualization of the whole body’s vasculature at a high resolution is demonstrated by the video included in the Appendix A. Scanning mode: step and shoot scanning, rotr. step 0.16°, averaging frames 2, filter Al 1 mm, magnification 16 µm. Software InstaRecon^®^ (Bruker) was used for 3D reconstructions; the morphological 2D/3D analysis was performed in CTan^®^ software, v.1.18 (Bruker).

**Figure 11 pharmaceuticals-17-01479-f011:**
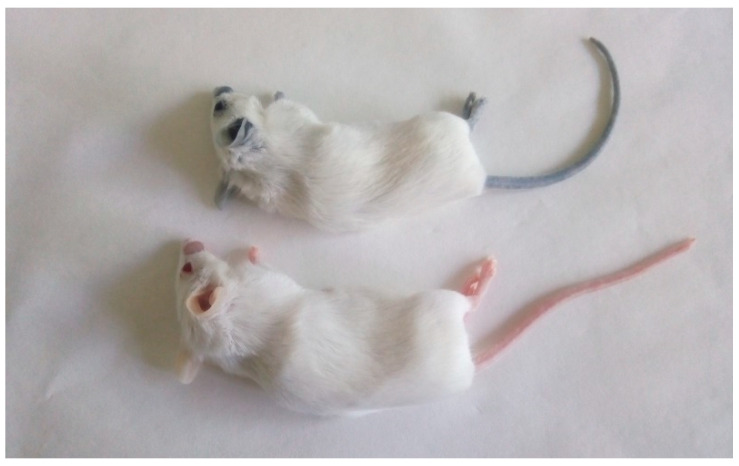
Deposition of AuNGs in the skin of mice. Mouse treated intra venously with nanogold particles at 20 mg/mice (**upper**). The apical parts of mice (tail, legs, ear, eyes and nose) turned blue immediately after intra venous administration and persisted during the whole life span of the animal (experiment lasted 9 months). Untreated mouse (**lower**). Sleeping mice after anesthesia.

**Figure 12 pharmaceuticals-17-01479-f012:**
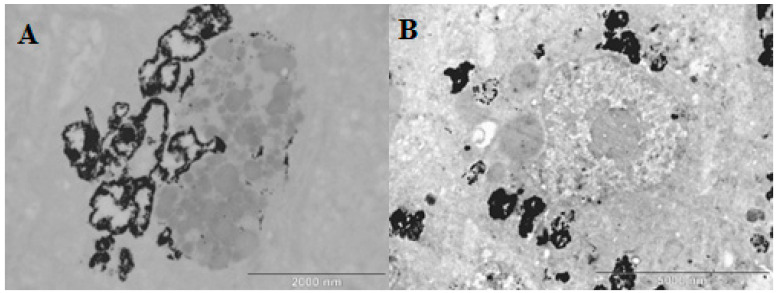
Deposition of nanogold particles in hepatocytes and spleen, visualized by TEM. (**A**) hepatocytes; (**B**) spleen macrophages.

**Figure 13 pharmaceuticals-17-01479-f013:**
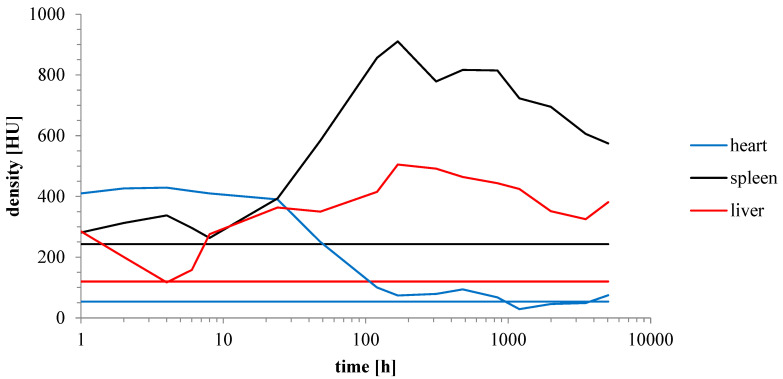
Radio density of heart, spleen, and liver of mice after dosed with PEGylated AuNPs; straight lines represent radio density of heart, spleen and liver of untreated mice.

## Data Availability

Data Availability Statements are available in section.

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
