# Peer review of "Preparation and Complex Characterisation of Stabilised Gold Nanoparticles: Biodistribution and Application for High Resolution In Vivo Imaging"

_pharmaceuticals, 2024, doi:10.3390/ph17111479_

Round 1

Reviewer 1 Report

Comments and Suggestions for Authors

In this work, the authors proposed PEG-stabilized gold nanoparticles for potential application in the micro-CT imaging. The manuscript evokes mixed impression, так как имеет сильные и significantly weak sides. The most important question is the following: what is a novelty of the proposed method in comparison with the well-known Turkevich method? There are many papers devoted to the post-modification of spherical and non-spherical gold NPs, what are the advantages and novelty of the author's method in comparison with them? The prepared gold nanoparticles undergo to comprehensive characterization by different techniques. On the one hand, it is positive that the data from different methods are generally consistent with each other. On the other hand, the characterization of well-known particles in itself does not carry novelty or scientific value. In my opinion, it is necessary to shift the study accents on the biological testing of the prepared nanoparticles. In the current version of the manuscript, a large part of key information relevant to the Pharmaceuticals scope remained in supporting information, while a large amount of data exclusively on the nanoparticles characterization is in the main text. In addition, the manuscript lacks a discussion of the obtained results in each part, while it is quite superficial. Further, the major issues which must be solved by the authors to the manuscript could pretend on to be acceptable.

In general:

1. Novelty of the research must be clearly postulated.

2. Discussion of the results in parts 3.9-3.12 should be expanded.

3. English writing, including the pointing should be revised. The manuscript in the current form contains a lot of typos, including in the sub-titles.

Introduction

4. Introduction is too short and should be expanded. Just eight references do not cover even briefly the current biomedical applications, properties and methods for synthesizing gold particles.

5. It is necessary to the references 3-6 in the second paragraph of introduction to specify for various applications of gold NPs.

6. Lines 61-63: the authors mention three different basic methods for synthesizing spherical NPs, but provide one reference. Separate references to original papers must be given for each method.

Material and Methods

7. There is no section describing the reagents used.

8. The final concentration after PEG-AuNPs centrifugation should be indicated (line 86).

Results and Discussion

9. The authors have characterized and shown how good citrate-stabilized NPs are, but this has long been known to them. Therefore, this entire part can be moved to supporting information. In the main text, only data should be left that reveals the comparison of the properties of the particles before and after modification.

10. The axis titles in right part of Figure 2 should be corrected.

11. Figure 4 requires revision; duplication of information should be eliminated.

12. The authors should give the details how they calculated the size and concentration of NPs from the UV-Vis spectra.

13. Interpretation of FTIR spectra is doubtful. The FTIR spectra of methoxy-PEG-thiol and citrate-stabilized NPs have to be included for comparison.

14. Illustrative data, supporting part 3.8. has be moved to the main text as it brings important information.

15. Error bars have to be added to Figure S2.

16. How can the authors explain the better contrast using their particles compared to the commercial sample? Since, only the gold core works in CT tomography?

17.  Part 3.11 titled as “Biodistribution of AuNP” does not provide the relevant information. The authors only note the accumulation of NPs in the skin, while the real distribution between the organs is not presented. Therefore, points 3.11 and 3.12 should be joined. If the experiment lasted 9 months, it would be good to show several photos made with periodicity at least 3 mounts. If the blueness of the skin persists for 9 months, does this mean that the NPs are not excreted from the body? In this case, can they be considered as CT-contrast at all? There is no image of a mouse that was injected with a commercial contrast agent.

Conclusions

18. Conclusions are too general and are more appropriate for introduction rather than for conclusions. So, conclusions have to be rewritten with a focus on exact outcomes obtained from the presented experimental data.

19. Repeated micro-CT scanning is postulated by the authors but no supporting data is presented in the manuscript (CT-images at different time points or something else). Authors should either provide such information or remove this mention from the manuscript.

Comments on the Quality of English Language

English writing, including the pointing should be revised. 

Author Response

In this work, the authors proposed PEG-stabilized gold nanoparticles for potential application in the micro-CT imaging. The manuscript evokes mixed impression, так как имеет сильные и significantly weak sides. The most important question is the following: what is a novelty of the proposed method in comparison with the well-known Turkevich method? There are many papers devoted to the post-modification of spherical and non-spherical gold NPs, what are the advantages and novelty of the author's method in comparison with them? The prepared gold nanoparticles undergo to comprehensive characterization by different techniques. On the one hand, it is positive that the data from different methods are generally consistent with each other. On the other hand, the characterization of well-known particles in itself does not carry novelty or scientific value. In my opinion, it is necessary to shift the study accents on the biological testing of the prepared nanoparticles. In the current version of the manuscript, a large part of key information relevant to the Pharmaceuticals scope remained in supporting information, while a large amount of data exclusively on the nanoparticles characterization is in the main text. In addition, the manuscript lacks a discussion of the obtained results in each part, while it is quite superficial. Further, the major issues which must be solved by the authors to the manuscript could pretend on to be acceptable.

Answer: A number of articles are published on the topic of preparing gold nanoparticles, which only deal with certain aspects of the problem. Commercial preparations are extremely expensive and do not allow larger experiments to be carried out within the framework of grants.

We have not found in any publication as extensive a comparison of methods and experimental data for the characterization of gold nanoparticle preparations as we used in our manuscript. It will be beneficial for the reader to compare the information that each method provides and what methods he can use given their availability in his laboratory. It is also evident that different gold nanoparticles, although of the same size, can provide different intensity of contrast depending on their internal structure. An optimized laboratory preparation can provide better contrast gold nanoparticles than commercial preparations of different batches. The article does not aim to bring groundbreaking information, but reasonable experimental data on the preparation and comprehensive analytical characterization of laboratory-prepared gold nanoparticles and their use in imaging the cardiovascular system using the high-resolution microcomputed tomography. How useful the work will be for readers will be shown only by the number of citations.

In general:

1. Novelty of the research must be clearly postulated.

The extensive a comparison of methods and solid experimental data on the preparation and comprehensive analytical characterization of laboratory-prepared gold nanoparticles and their use in imaging the cardiovascular system in small laboratory animals using the high-resolution microcomputed tomography.

  1. Discussion of the results in parts 3.9-3.12 should be expanded.

Corrected. It is surprising that gold nanoparticles of similar size and size distribution differed so much in the contrast provided. The reason may be their half-life in the bloodstream due to the stability of their surface coverage. Also, the internal structure of their core given by the method and execution of the preparation can have an effect on the contrast intensity. Our experimental work was not focused in this direction, and to clarify the influence of the internal structure of gold nanoparticles on the contrast intensity, a new grant project needs to be prepared.

Persistent deposition of gold nanoparticles in the skin and eyes and bluing of the skin should be taken into account when performing some in vivo studies, as such studies may significantly affect.

Deposition of the tested gold nanoparticles in the skin and other internal organs had no observable toxic effect on mice. Which is consistent with the inertness of gold nanoparticles and their low pro-inflammatory potential, as shown by in vitro experiments. Nevertheless, the accumulation of gold nanoparticles (larger than 8 nm) should be considered as a possible source of interference in pharmacological experiments where computed microtomography is used.

  1. English writing, including the pointing should be revised. The
    manuscript in the current form contains a lot of typos, including in the
    sub-titles.

Answer: The text of manuscript was checked by native speaking  colleague and corrected where appropriate. 

Introduction

4. Introduction is too short and should be expanded. Just eight
references do not cover even briefly the current biomedical
applications, properties and methods for synthesizing gold particles.

Corrected.

5. It is necessary to the references 3-6 in the second paragraph of
introduction to specify for various applications of gold NPs.
Lines 61-63: the authors mention three different basic methods for
synthesizing spherical NPs, but provide one reference. Separate
.references to original papers must be given for each method.

Corrected. 

Material and Methods

7. There is no section describing the reagents used.
Answer: Reagents used are described in each paragraph in the section Material and Methods as it is standard way.
8. The final concentration after PEG-AuNPs centrifugation should be
indicated (line 86).

Answer: Ultracentrifugation I used only to remove nanogold from reaction mixture and in consequent washing step. Concentration of gold nanoparticles I done in finally diluted preparation.

Results and Discussion

9. The authors have characterized and shown how good citrate-stabilized
NPs are, but this has long been known to them. Therefore, this entire
part can be moved to supporting information. In the main text, only data
should be left that reveals the comparison of the properties of the
particles before and after modification.

It has been modified accordingly
10. The axis titles in right part of Figure 2 should be corrected.
Corrected

  1. Figure 4 requires revision; duplication of information should be
    eliminated.
    Corrected

  2. The authors should give the details how they calculated the size and
    concentration of NPs from the UV-Vis spectra.
    Calculation was done exactly according the method described in citation 15. Report. Using uv-vis as a tool to determine size and concentration of spherical gold nanoparticles (sgnps) from 5 to 100 nm; 2008; pp 1-3.
    13. Interpretation of FTIR spectra is doubtful. The FTIR spectra of
    methoxy-PEG-thiol and citrate-stabilized NPs have to be included for
    comparison.
    Answer: . The FTIR spectra of methoxy-PEG-thiol and citrate-stabilized NPs were used as published in literature and were used to identify peaks in Fig. 6. Peak of Au-S vibration was identified according the data published in reference by Omar at al. Omar, N.A.S.; Fen, Y.W.; Abdullah, J.; Kamil, Y.M.; Daniyal, W.; Sadrolhosseini, A.R.; Mahdi, M.A. Sensitive detection of dengue virus type 2 e-proteins signals using self-assembled monolayers/reduced graphene oxide-pamam dendrimer thin film-spr optical sensor. Scientific Reports 2020, 10.
    14. Illustrative data, supporting part 3.8. has be moved to the main
    text as it brings important information.
    Answer: Accepted and done
    15. Error bars have to be added to Figure S2.
    Corrected
    16. How can the authors explain the better contrast using their
    particles compared to the commercial sample? Since, only the gold core
    works in CT tomography?
    Some explanation was added to the section 3.10. It is surprising that gold nanoparticles of similar size and size distribution differed so much in the contrast provided. The reason may be their half-life in the bloodstream due to the stability of their surface coverage. Also, the internal structure of their core given by the method and execution of the preparation can have an effect on the contrast intensity. Our experimental work was not focused in this direction, and to clarify the influence of the internal structure of gold nanoparticles on the contrast intensity, a new grant project needs to be prepared
    17. Part 3.11 titled as “Biodistribution of AuNP” does not provide the
    relevant information. The authors only note the accumulation of NPs in
    the skin, while the real distribution between the organs is not
    presented. Therefore, points 3.11 and 3.12 should be joined. If the
    experiment lasted 9 months, it would be good to show several photos made
    with periodicity at least 3 mounts. If the blueness of the skin persists
    for 9 months, does this mean that the NPs are not excreted from the
    body? In this case, can they be considered as CT-contrast at all? There
    is no image of a mouse that was injected with a commercial contrast
    agent.
    Answer: Gold nanoparticles of size obove 8 nm are not excreted from the body. That is the reason why this gold nanoparticles are use in preclinical research only.

Figure 13. Radio density in time of heart, spleen, and liver of mice after PEGylated AuNPs dosing, straight lines represent radio density of heart, spleen and liver of untreated mice.

Conclusions

18. Conclusions are too general and are more appropriate for
introduction rather than for conclusions. So, conclusions have to be
rewritten with a focus on exact outcomes obtained from the presented
experimental data.
Corrected.

  1. Repeated micro-CT scanning is postulated by the authors but no
    supporting data is presented in the manuscript (CT-images at different
    time points or something else). Authors should either provide such
    information or remove this mention from the manuscript.
    Answer: In Fig.13 is clearly shown that scanning experiment continued in time frame 5000 hours it is about 208 days (5000 : 24 ꞊ 208)  it means about 7 months (208 : 30 = 6.93). observation of  mice skin colour lasted 9 months as it is written in text.

Comments on the Quality of English Language
English writing, including the pointing should be revised.
Answer: English was revised, typing errors were corrected

Reviewer 2 Report

Comments and Suggestions for Authors

The authors in the manuscript entitled "Preparation and complex characterisation of stabilised gold nanoparticles: study on biodistribution and application for in vivo imaging by µCT" did an interesting work.

The title is not making a sense of some influential work which need to be rephrased.

There are so many reports on stabilized NPs, even with PEG polymer with very good results which should be discussed in manuscript but the authors even did not cite such as (few not all)

1.      Wang, Y., Quinsaat, J.E.Q., Ono, T. et al. Enhanced dispersion stability of gold nanoparticles by the physisorption of cyclic poly(ethylene glycol). Nat Commun 11, 6089 (2020). https://doi.org/10.1038/s41467-020-19947-8

2.      K. Bharti, M. A. Sk and K. K. Sadhu, Seed free synthesis of polyethylene glycol stabilized gold nanoprisms exploiting manganese metal at low pH, Nanoscale Advances, (2023), 5(14), 3729-3736.

How the authors claim novelty on this part of Fabricated PEG@AuNPs?

The author in title emphasized on the use of µCT, even they also have used many other instruments to assess the nanoparticle; my concern is that what is the novelty in work to use µCT and emphasizing in title? A step advanced work entitledFabrication of targeted gold nanoparticle as potential contrast agent in molecular CT imaging” has been reported in 2023 (https://doi.org/10.1016/j.jrras.2022.100490) with very good results (has not even been cited by authors); how the authors justify the superiority of their work over this work?  Over all the title does not address a specific gap in the field but more likely it reflects a trivial work.

material and method section lacks reader friendly protocols; somewhat tedious and lack of proper designing, for example the authors didn’t explain what type of mouse model they prepared and using what type of protocol and how many mouse they were used to study? how they induce cancer in mouse? The mouse model study, in brief, totally look shallow. In viability test the authors did not mention; how much cells were taken to incubate using one concentration of PEG-AuNP? what are the incubation conditions? And similarly other protocols were described with lot of flaws.

In results section; the characterization of fabricated material not performed properly. The authors didn’t performed X-ray, SEM, analysis. The TEM analysis not give clear reflection of particle formation. Therefore, there is a need to repeat whole work by superseding the work already published.  

As the work bearing many flaws, therefore, need to reshape the conclusion after performing whole study again as I suggested above.

The authors left uncited many references which are most related to this study; as shown few above as an example.

Author Response

The title is not making a sense of some influential work which need to
be rephrased.

Corrected.

There are so many reports on stabilized NPs, even with PEG polymer with
very good results which should be discussed in manuscript but the
authors even did not cite such as (few not all)

1. Wang, Y., Quinsaat, J.E.Q., Ono, T. et al. Enhanced dispersion
stability of gold nanoparticles by the physisorption of cyclic
poly(ethylene glycol). Nat Commun 11, 6089 (2020).
https://doi.org/…7-8

2. K. Bharti, M. A. Sk and K. K. Sadhu, Seed free synthesis of
polyethylene glycol stabilized gold nanoprisms exploiting manganese
metal at low pH, Nanoscale Advances, (2023), 5(14), 3729-3736.
There is a lot of articles on metallic nanoparticles. Haw is manganese metal relevant to nanogold?

Corrected.
How the authors claim novelty on this part of Fabricated PEG@AuNPs?

The author in title emphasized on the use of µCT, even they also have
used many other instruments to assess the nanoparticle; my concern is
that what is the novelty in work to use µCT and emphasizing in title? A
step advanced work entitled “Fabrication of targeted gold nanoparticle
as potential contrast agent in molecular CT imaging” has been reported
in 2023 (https://doi.org/…490) with very good
results (has not even been cited by authors); how the authors justify
the superiority of their work over this work? Over all the title does
not address a specific gap in the field but more likely it reflects a
trivial work.
material and method section lacks reader friendly protocols; somewhat
tedious and lack of proper designing, for example the authors didn’t
explain what type of mouse model they prepared and using what type of
protocol and how many mouse they were used to study? how they induce
cancer in mouse? The mouse model study, in brief, totally look shallow.

Answer: There are no cancer models used.

In viability test the authors did not mention; how much cells were taken
to incubate using one concentration of PEG-AuNP? what are the incubation
conditions? And similarly other protocols were described with lot of
flaws.
Answer: The mentioned paper do not deal with 15 nm stabilized AuNP, but with PEG-Fe3O4@Au NPs of about 41.5nm. There are no data on in vivo imaging.

Detailed protocols are a part of commercial tests used in our study.   

In results section; the characterization of fabricated material not
performed properly. The authors didn’t performed X-ray, SEM, analysis.
The TEM analysis not give clear reflection of particle formation.
Therefore, there is a need to repeat whole work by superseding the work
already published.

As the work bearing many flaws, therefore, need to reshape the
conclusion after performing whole study again as I suggested above.

The authors left uncited many references which are most related to this
study; as shown few above as an example.

Corrected. 

Round 2

Reviewer 1 Report

Comments and Suggestions for Authors

The authors addressed partially to the given comments.